# TRAINING-FREE MULTI-OBJECTIVE DIFFUSION MODEL FOR 3D MOLECULE GENERATION

**Xu Han**[1], **Caihua Shan**[2], **Yifei Shen**[2], **Can Xu**[3], **Han Yang**[4], **Xiang Li**[3], **Dongsheng Li**[2]
[1]Tufts University, [2]Microsoft Research Asia
[3]East China Normal University, [4]Microsoft Research AI4Science
Xu.Han@tufts.edu, leoxc1571@163.com, xiangli@dase.ecnu.edu.cn
{caihuashan, yifeishen, hanyang, dongsheng.li}@microsoft.com

## ABSTRACT

Searching for novel and diverse molecular candidates is a critical undertaking in drug and material discovery. Existing approaches have successfully adapted the diffusion model, the most effective generative model in image generation, to create 1D SMILES strings, 2D chemical graphs, or 3D molecular conformers. However, these methods are not efficient and flexible enough to generate 3D molecules with multiple desired properties, as they require additional training for the models for each new property or even a new combination of existing properties. Moreover, some properties may potentially conflict, making it impossible to find a molecule that satisfies all of them simultaneously. To address these challenges, we present a training-free conditional 3D molecular generation algorithm based on off-the-shelf unconditional diffusion models and property prediction models. The key techniques include modeling the loss of property prediction models as energy functions, considering the property relation between multiple conditions as a probabilistic graph, and developing a stable posterior estimation for computing the conditional score function. We conducted experiments on both single-objective and multi-objective 3D molecule generation, focusing on quantum properties, and compared our approach with the trained or fine-tuned diffusion models. Our proposed model achieves superior performance in generating molecules that meet the conditions, without any additional training cost.

## 1 INTRODUCTION

Diffusion models have emerged as a powerful family of deep generative models across various domains, including image generation, audio synthesis, etc. Due to their ability to generate novel and diverse molecule structures, they help accelerate the process of drug discovery by reducing the need for expensive and time-consuming wet experiments (Schwalbe-Koda & Gómez-Bombarelli, 2020).

Here we focus on 3D molecules, as 3D structures provide a more accurate representation of the spatial arrangement of atoms and molecule's geometric symmetry in comparison to 2D graph representation. This enhanced accuracy results in a better generalization (Thomas et al., 2018; Fuchs et al., 2020; Finzi et al., 2020). Moreover, 3D molecule design enables the identification of promising drug candidates with a range of characteristics, such as quantum properties (Ramakrishnan et al., 2014), chirality (Adams et al., 2021), binding affinity with specific proteins (Lin et al., 2022), etc.

Given its practical importance, there have been some pioneering works on conditional 3D molecule generation (Hoogeboom et al., 2022; Xu et al., 2023; Bao et al., 2022). For instance, EDM (Hoogeboom et al., 2022) is the first proposed diffusion model for 3D molecules, which learns an equivariant neural network that jointly operates on both atom coordinates and atom types. Despite good performance in generating molecules with a single desired property, the flexibility and efficiency are limited. First, additional training is required to incorporate conditional guidance. EDM and GEOLDM (Xu et al., 2023) directly retrain a new diffusion model conditioned on a particular prop-

---

This work was done during Xu Han's internship at MSRA. Correspondence to: Caihua Shan (caihuashan@microsoft.com; Dongsheng Li (dongsheng.li@microsoft.com).

erty. EEGSDE (Bao et al., 2022) trains a time-dependent energy function to perform the guidance. Substantial efforts should be paid to collect the dataset and train models, which is non-negligible. Additionally, training a time-dependent energy function is a difficult task, as the molecules at the early stages of diffusion are noisy and even invalid. Second, to accommodate multiple conditions, all previous works need to retrain the model for every combination of conditions, resulting in unaffordable costs. Third, the given properties might conflict, making it hard to find a molecule that meets all conditions. Under such circumstances, existing methods also lack sufficient flexibility to achieve the best balance among multiple properties, further limiting their effectiveness.

To address the above challenges, we propose MuDM, a training-free multi-objective 3D molecule generation based on off-the-shelf pre-trained diffusion model and property prediction functions. Considering the intermediate molecules $\mathbf{z}_t$ are noisy, we map it to the expected final molecule $\hat{\mathbf{z}}_0$ by posterior approximation, use Monte Carlo sampling to compute the gradient of the property functions with low variance, and map back to update $\mathbf{z}_{t-1}$ by chain rule (Chung et al., 2022; Song et al., 2023). To tackle the multiple conditions, we utilize the probabilistic graph to model the complex relationships of properties, and adjust the expected final molecule $\hat{\mathbf{z}}_0$ based on the dependency of properties in the graph. The overall conditional guidance is then calculated as the weighted sum of the gradient of each property with respect to the expected revised final molecule $\mathbf{z}_0'$. The revised $\mathbf{z}_0'$ and gradient weights potentially identify the importance of properties and mitigate the conflict among given conditions.

We conducted the experiments on both single-objective and multi-objective tasks, aiming to generate 3D molecules with specific quantum properties. Compared with training-required methods, our proposed training-free MuDM achieves superior performance in all cases for single-objective tasks. Furthermore, MuDM is capable of generating 3D molecules with any combination of properties, resulting in an average reduction of $\sim 30\%$ in mean square error for each property.

Our contributions can be summarized as follows:

1. By incorporating posterior approximation and MC sampling, MuDM can provide accurate guidance that depends solely on off-the-shelf time-independent property functions.

2. MuDM is capable of managing complex relationships among properties as a probabilistic graph, and then deriving the effective approximation for the multi-objective guidance.

3. Experimental results reveal that MuDM is an effective and efficient method to design 3D molecules with desired properties, offering great flexibility in exploring the chemical space.

## 2 RELATED WORK

**Diffusion Models.**  Diffusion models (Ho et al., 2020; Song et al., 2020) generate samples by modeling the generative process as the reverse of the noising process. Specifically, the noising process injects the noise into the ground truth data. The generative process learns the score functions, which is the gradient of the ground truth data distribution's log-density, to reverse the noising process.

Conditional diffusion models enable the generative process with constraints, where the score function should be learned not only from the ground truth data distribution but also from the given constraints. They can be broadly categorized into two types: training-required and training-free. A well-known example of training-required methods is Stable Diffusion (Rombach et al., 2022) for text-guided image generations. However, the extra training cost can be significant, particularly in scenarios that require complex control with multiple conditions. Consequently, our paper focuses on exploring training-free methods for their efficiency and flexibility in handling diverse constraints.

**Molecule Generation with Desired Properties.**  To the best of our knowledge, current approaches for conditional 3D molecule generation mostly utilize the condition diffusion models. EDM (Hoogeboom et al., 2022) and GEOLDM (Xu et al., 2023) trained a specific diffusion model for each property. EEGSDE (Bao et al., 2022) is more efficient to train a time-dependent property function instead of re-training a new diffusion model, and then guide the generative process. In contrast, our proposed method is training-free, which only requires the off-the-shelf pre-trained diffusion model and time-independent prediction functions, making it the most efficient option. Furthermore, the quality of generated 3D molecules is also confirmed by the experiments.

Other traditional techniques, such as reinforcement learning (De Cao & Kipf, 2018; You et al., 2018; Popova et al., 2018; Shi et al., 2020) and genetic algorithms (Jensen, 2019; Ahn et al., 2020; Nigam et al., 2019), have been applied to generate molecules with specific properties. Additionally, recent studies have also tackled multi-objective molecule problems, as seen in (Jin et al., 2020; Xie et al., 2021). However, they all consider 2D graphs rather than 3D conformers, posing challenges in directly applying their techniques to 3D molecule generation.

## 3   BACKGROUND

**3D Molecule Representation.** Let us consider a molecule consisting of $N$ atoms. For the $i^{th}$ atom, its coordinate can be represented as $\mathbf{x}_i \in \mathbb{R}^3$, denoting the three-dimensional space. Collectively, the coordinates for all atoms can be represented as $\mathbf{x} = (\mathbf{x}_1, \ldots, \mathbf{x}_N) \in \mathbb{R}^{N \times 3}$, which encapsulates the molecule's conformation. Beyond the spatial coordinates, every atom also has specific features, such as atomic type. Let $\mathbf{h}_i \in \mathbb{R}^d$ symbolize the feature vector for the $i^{th}$ atom. Aggregating the features for all atoms, we obtain $\mathbf{h} = (\mathbf{h}_1, \ldots, \mathbf{h}_N) \in \mathbb{R}^{N \times d}$. Consequently, a molecule can be represented as $\mathcal{G} = [\mathbf{x}, \mathbf{h}]$, integrating both its three-dimensional geometry and atomic features.

**Invariance.** Two fundamental properties associated with 3D molecules are invariance and equivariance. Given a transformation $R$, a distribution $p(\mathbf{x}, \mathbf{h})$ is said to be invariant to $R$ if: $p(\mathbf{x}, \mathbf{h}) = p(R\mathbf{x}, \mathbf{h})$ for all $\mathbf{x}$ and $\mathbf{h}$. Here $R\mathbf{x} = (R\mathbf{x}_1, ..., R\mathbf{x}_M)$ denotes the transformation applied to each coordinate. Further, a function $f(\mathbf{x}, \mathbf{h})$ is invariant to $R$ if: $f(R\mathbf{x}, \mathbf{h}) = f(\mathbf{x}, \mathbf{h})$ for all $\mathbf{x}$ and $\mathbf{h}$. Invariance implies that the transformation $R$ does not affect the outcome.

**Equivariance.** Consider a function $f$ with outputs $a_\mathbf{x}$ and $a_\mathbf{h}$, i.e., $(a_\mathbf{x}, a_\mathbf{h}) = f(\mathbf{x}, \mathbf{h})$. This function is said to be equivariant to $R$ if: $f(R\mathbf{x}, \mathbf{h}) = (Ra_\mathbf{x}, a_\mathbf{h})$ for all $\mathbf{x}$ and $\mathbf{h}$. Equivariance means that changing the order of $f$ and $R$ does not affect the outcome.

**Equivariant diffusion Model (EDM) and Geometric Latent Diffusion Models (GEOLDM).** They are both diffusion models for 3D molecules, which are built upon the principles proposed by (Ho et al., 2020; Kingma et al., 2021), but with different spaces to conduct the noising and generative process. The nosing process of EDM is defined in the original molecule space $\mathcal{G} = [\mathbf{x}, \mathbf{h}]$:

$$q(\mathcal{G}_{1:T}|\mathcal{G}) = \prod_{t=1}^{T} q(\mathcal{G}_t|\mathcal{G}_{t-1}), \quad q(\mathcal{G}_t|\mathcal{G}_{t-1}) = \mathcal{N}_{xh}(\mathcal{G}_t; \sqrt{\alpha_t}\mathcal{G}_{t-1}, \beta_t I)$$

where $\mathcal{N}_{xh}(\mathcal{G}_t; \sqrt{\alpha_t}\mathcal{G}_{t-1}, \beta_t I) = \mathcal{N}_X(\mathbf{x}_t; \sqrt{\alpha_t}\mathbf{x}_{t-1}, \beta_t I)\mathcal{N}(\mathbf{h}_t; \sqrt{\alpha_t}\mathbf{h}_{t-1}, \beta_t I)$. EDM adds the noise into atom coordinates and atom features separately, where $\alpha_t + \beta_t = 1$ and they both control the noise scale. $\mathcal{N}_X$ is the Gaussian distribution of the coordinates in the zero CoM subspace to guarantee the translational invariance. Accordingly, the reverse generative process of EDM is:

$$p(\mathcal{G}_{1:T}|\mathcal{G}) = p(\mathcal{G}_T)\prod_{t=1}^{T} p(\mathcal{G}_{t-1}|\mathcal{G}_t), \quad p(\mathcal{G}_{t-1}|\mathcal{G}_t) = \mathcal{N}_X(\mathbf{x}_{t-1}; \mu_{\theta_1}(\mathbf{x}_t), \tilde{\beta}_t I)\mathcal{N}(\mathbf{h}_{t-1}; \mu_{\theta_2}(\mathbf{h}_t), \tilde{\beta}_t I).$$

Different from EDM, GEOLDM is to capture the 3D molecule's equivariance and invariance constraints in the latent space. GEOLDM learns the encoder $\mathbf{z} = [\mathbf{z}_x, \mathbf{z}_h] = \mathcal{E}_\phi(\mathbf{x}, \mathbf{h})$ and the decoder $\mathbf{x}, \mathbf{h} = \mathcal{D}_\xi(\mathbf{z})$, where $\mathcal{E}_\phi$ and $\mathcal{D}_\xi$ are both equivariant graph neural networks (EGNNs, Satorras et al. (2021)). For all the rotations $R$ and translations $\mathbf{t}$, they satisfy

$$R\mathbf{z}_x + \mathbf{t}, \mathbf{z}_h = \mathcal{E}_\phi(R\mathbf{x} + \mathbf{t}, \mathbf{h}), \quad R\mathbf{x} + \mathbf{t}, \mathbf{h} = \mathcal{D}_\xi(R\mathbf{z}_x + \mathbf{t}, \mathbf{z}_h), \quad p(\mathbf{z}_x, \mathbf{z}_h) = p(R\mathbf{z}_x, \mathbf{z}_h). \quad (1)$$

The diffusion process of GEOLDM is also defined on the latent space. Its noising process and reverse generative process are similar to EDM, where $\mathbf{z}_t = [\mathbf{z}_{x,t}, \mathbf{z}_{h,t}]$:

$$q(\mathbf{z}_t|\mathbf{z}_{t-1}) = \mathcal{N}(\mathbf{z}_t; \alpha_t\mathbf{z}_{t-1}, \beta_t I), \quad p_\theta(\mathbf{z}_{t-1}|\mathbf{z}_t) = \mathcal{N}(\mathbf{z}_{t-1}; \mu_\theta(\mathbf{z}_t, t), \sigma_t^2 I).$$

The optimization objective of GEOLDM is:

$$\mu_\theta(\mathbf{z}_t, t) = \frac{1}{\sqrt{\alpha_t}}(\mathbf{z}_t - \frac{\beta_t}{\sqrt{1-\bar{\alpha}_t}}\epsilon_\theta(\mathbf{z}_t, t)), \quad \mathbb{E}_{\mathcal{E}(\mathcal{G}), \epsilon \sim \mathcal{N}(0,I), t}[||\epsilon - \epsilon_\theta(\mathbf{z}_t, t)||^2].$$

**Inverse problems.** Conditional diffusion models have shown great potential in solving inverse problems. Suppose we have a measurement $y \in \mathbb{R}$ from the forward measurement operator $\mathcal{A}(.)$,

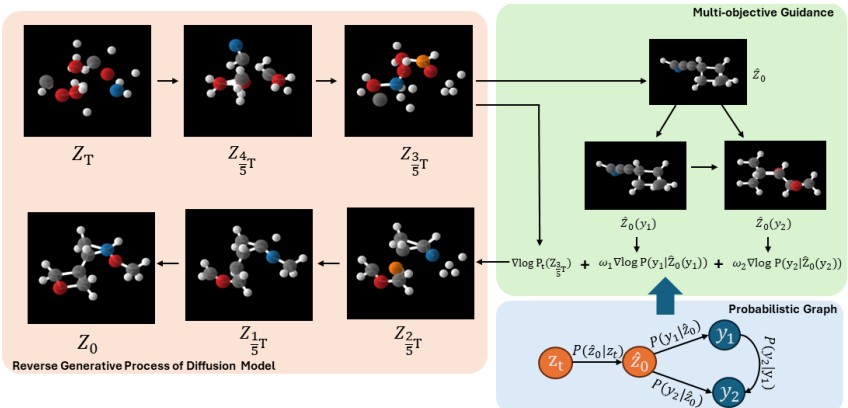

Figure 1: Overview: MuDM generates a molecule from the time $T$ to $0$ using a pre-trained diffusion model, guided by multiple property functions. The relationship of multiple properties is modeled by a given probabilistic graph. At some time step, the guidance is computed as the weighted sum of the gradient of each property with respect to the latent variable considering the property dependencies.

which satisfies $y = \mathcal{A}(\mathbf{x}) + n$ where $n$ represents gaussian noise. Given the observed measurement $y$, diffusion models can be used to retrieve $\mathbf{x}$ by replacing the score function $\nabla_{\mathbf{x}_t} \log p_t(\mathbf{x}_t)$ with the conditional score function $\nabla_{\mathbf{x}_t} \log p_t(\mathbf{x}_t|y)$. However, the absence of an analytical formulation for $\log p_t(y|\mathbf{x}_t)$ poses a significant challenge, leading to many studies (Chung et al., 2022; Song et al., 2022) proposing various approximations to address this issue.

## 4 METHODOLOGY

In this section, we treat conditional 3D molecule generation as an inverse problem and develop a feasible solution shown in Fig. 1. Traditionally, molecular design has been a forward process, where researchers synthesize and test molecules to observe their properties. Inverse molecular design, in contrast, reverses the process by starting with the desired properties and property functions, and working backward to identify molecules that exhibit those characteristics. We will first discuss how to generate a molecule with a single property, and delve into multi-objective tasks in the next section.

### 4.1 SINGLE-CONDITION GUIDANCE

Suppose we have a property predictor $\mathcal{A}_\psi(.) : \mathcal{G} \to \mathbb{R}$, the target property value $y \in \mathbb{R}$, and a loss function $\ell : \mathbb{R} \times \mathbb{R} \to \mathbb{R}$. We define the likelihood function as

$$p(y|\mathcal{G} = \mathcal{D}(\mathbf{z}_0)) \propto \exp(-\ell(\mathcal{A}_\psi(G), y)). \tag{2}$$

To recover the molecules' prior distribution starting from the tractable distribution, the generative process of diffusion models can be also expressed as a reverse SDE:

$$d\mathbf{z} = \left[ -\frac{\beta_t}{2}\mathbf{z} - \beta_t \nabla_{\mathbf{z}_t} \log p_t(\mathbf{z}_t) \right] dt + \sqrt{\beta_t} d\bar{\mathbf{w}}. \tag{3}$$

Now we will discuss how to incorporate the desired property into the molecule generation process. The mapping from the property to the molecule $y \to \mathcal{G}$ is one-to-many, which is an ill-posed inverse problem. From the Bayesian point, we treat $p(\mathbf{z}_0)$ as the prior, and samples from the posterior $p(\mathbf{z}_0|y)$. Based on Bayes' rule, we know the fact:

$$\nabla_{\mathbf{z}_t} \log p_t(\mathbf{z}_t|y) = \nabla_{\mathbf{z}_t} \log p_t(\mathbf{z}_t) + \nabla_{\mathbf{z}_t} \log p_t(y|\mathbf{z}_t). \tag{4}$$

Then Equation (3) can be modified as:

$$d\mathbf{z} = \left[ -\frac{\beta_t}{2}\mathbf{z} - \beta_t(\nabla_{\mathbf{z}_t} \log p_t(\mathbf{z}_t) + \nabla_{\mathbf{z}_t} \log p_t(y|\mathbf{z}_t)) \right] dt + \sqrt{\beta_t} d\bar{\mathbf{w}}. \tag{5}$$

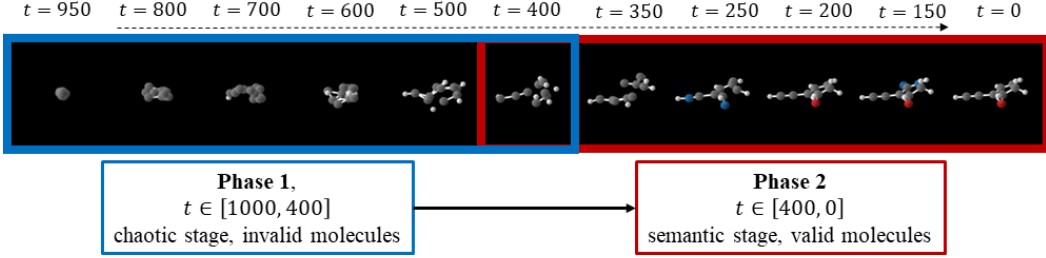

Figure 2: Demonstration of expected final molecules $\mathcal{G} = \mathcal{D}(\hat{\mathbf{z}}_0(\mathbf{z}_t))$ at different stages

Unfortunately, the second term $p_t(y|\mathbf{z}_t)$ is intractable in practice. In the previous work for 3D molecule generation, there are two main methods to overcome this. EDM (Hoogeboom et al., 2022) and GEOLDM (Xu et al., 2023) trained denoising model jointly with condition $\nabla_{\mathbf{z}_t} \log p_t(\mathbf{z}_t, y)$. Alternatively, EEGSDE (Bao et al., 2022) trained a model to predict the property directly from the noised latent variables $\log p_t(y|\mathbf{z}_t)$. As we said before, they are not efficient and flexible.

Instead, we propose an effective plug-and-play framework for controllable 3D molecule generation without any extra training. We demonstrate that only with a pre-trained diffusion model on the dataset $\nabla_{\mathbf{z}_t} \log p_t(\mathbf{z}_t)$ and the measurement operator $\mathcal{A}(\mathcal{G})$, we can generate molecules with high-quality and the desired property. We first rewrite the second term $p_t(y|\mathbf{z}_t)$ as:

$$p_t(y|\mathbf{z}_t) = \int p_t(y|\mathbf{z}_0, \mathbf{z}_t) p_t(\mathbf{z}_0|\mathbf{z}_t) d\mathbf{z}_0 = \int p_t(y|\mathbf{z}_0) p_t(\mathbf{z}_0|\mathbf{z}_t) d\mathbf{z}_0 = \mathbb{E}_{\mathbf{z}_0 \sim p(\mathbf{z}_0|\mathbf{z}_t)}[p(y|\mathcal{G} = \mathcal{D}(\mathbf{z}_0))]$$
(6)

Here $p(\mathbf{z}_0|\mathbf{z}_t)$ is still intractable. Diffusion Posterior Sampling (DPS, Chung et al. (2022)) is proposed to approximate Equation (6) as:

$$p_t(y|\mathbf{z}_t) \simeq p(y|\mathcal{G} = \mathcal{D}(\hat{\mathbf{z}}_0)) \text{ where } \hat{\mathbf{z}}_0 := \mathbb{E}_{\mathbf{z}_0 \sim p(\mathbf{z}_0|\mathbf{z}_t)}[\mathbf{z}_0] = \frac{1}{\sqrt{\bar{\alpha}_t}}(\mathbf{z}_t + (1 - \bar{\alpha}_t)\nabla_{\mathbf{z}_t} \log p_t(\mathbf{z}_t)).$$
(7)

DPS used Tweedies' formula (Efron, 2011; Kim & Ye, 2021) to compute the posterior mean $\hat{\mathbf{z}}_0$, which is useful when the amount of paired data $(\mathcal{G}, y)$ is limited.

However, we cannot generate good molecules with the desired property due to the limitation of DPS. DPS replaces $p_t(\mathbf{z}_0|\mathbf{z}_t)$ with a delta distribution around $\hat{\mathbf{z}}_0$. The work (Song et al., 2023) indicates such "point estimation" could be highly inaccurate. This issue is even more serious in the 3D molecule generation due to inaccurate prediction of $\hat{\mathbf{z}}_0$ in the early steps. Unlike other data formats such as images, molecule property is very sensitive to the atom's positions. We show the entire process of expected final molecules $\hat{\mathbf{z}}_0$ in Figure 2. There are two obvious phases: the chaotic stage, where the samples are far from valid molecules, and the semantic stage, where they can be modified as valid molecules. The atoms of $\mathcal{G} = \mathcal{D}(\hat{\mathbf{z}}_0)$ in the chaotic stage are not reliable, and the measurement $\mathcal{A}(.)$ can not handle such invalid molecules, leading to unreasonable guidance.

The boundary between the chaotic stage and the semantic stage is decided in two ways. Initially, we visualize the expected final molecules and find the time the molecules become widely distributed and have certain shapes. Secondly, we investigate the performance change in Appendix A.4.1 and determine that starting from 400 steps is efficient for the task. Therefore, we skip guidance in $[1000, 400]$, which is in the chaotic stage.

In addition to stage division, we implement three strategies to further ensure the stability of guidance: Monte-Carlo sampling, property checker and time-travel. Unlike DPS using delta distribution to approximate $p_t(y|\mathbf{z}_t)$, we choose $q(\mathbf{z}_0|\mathbf{z}_t) = \mathcal{N}(\hat{\mathbf{z}}_0, r_t^2 \mathbf{I})$, which is a Gaussian with mean being the MMSE estimate $\hat{\mathbf{z}}_0$ and the convariance is a hyperparameter $r_t$. Previous work (Song et al., 2022) also adopts MC sampling, but other problem settings are different. With this new approximation, we estimate:

$$\nabla \log p_t(y|\mathbf{z}_t) = \nabla \log \mathbb{E}_{\mathbf{z}_0 \sim p(\mathbf{z}_0|\mathbf{z}_t)}[p(y|\mathcal{G} = \mathcal{D}(\mathbf{z}_0))]$$

$$\approx \nabla \log \mathbb{E}_{\mathbf{z}_0 \sim q(\mathbf{z}_0|\mathbf{z}_t)}[p(y|\mathcal{G} = \mathcal{D}(\mathbf{z}_0))] = \nabla \log \left(\frac{1}{m}\sum_{i=1}^{m} \exp\left(-\ell\left(\mathcal{A}\left(\mathcal{D}(\mathbf{z}_0^i)\right), y\right)\right)\right)$$
(8)

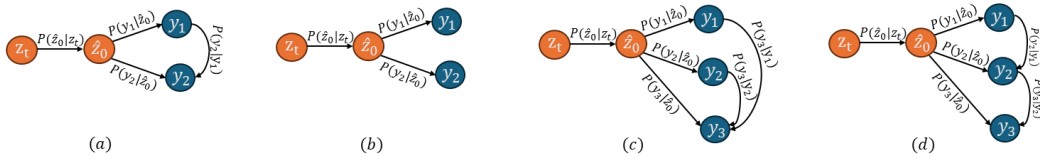

Figure 3: Different relationships of properties

where $\mathbf{z}_0^i$ is the i.i.d. sample from the distribution $q(\mathbf{z}_0|\mathbf{z}_t)$, and $m$ is the number of samples.

The property checker is to identify whether the guidance can be applied at the current step. We could first compute the range of the targeted property $[y_{min}, y_{max}]$ from the training dataset. At each step, we check if the estimation for the property is in this range $\mathcal{A}(\mathcal{D}(\mathbf{z}_0^i)) \in [y_{min}, y_{max}]$. Considering the invalid molecule can cause a very large estimation and hence influence the guidance direction, we opt not to apply any guidance at this step.

We employ a "time-travel" technique (Wang et al., 2022) to enhance the estimation of $\mathbf{z}_0$. This technique iteratively revisits prior steps in the denoising process, allowing for repeated refinement of the molecule generation. Specifically, after each denoising step that progresses the estimation from $\mathbf{z}_t$ to $\mathbf{z}_{t-1}$, we periodically revert to a previous step (e.g., from $\mathbf{z}_{t-1}$ back to $\mathbf{z}_t$) and reapply the denoising operation. This repeated denoising process serves to refine the estimation, ensuring a more accurate and stable convergence towards the desired $\mathbf{z}_0$.

We prove that our computed conditional score function is orthogonal equivariant if the loss function of the property is orthogonal invariant. The detailed proof is provided in Appendix A.2.2.

**Proposition 1.** *Suppose the loss function of the property $f = \ell(\mathcal{A}(\mathcal{D}(\cdot)), y)$ is invariant, i.e., $f(\mathbf{R}\mathbf{z}_{x,t}, \mathbf{z}_{h,t}) = f(\mathbf{z}_{x,t}, \mathbf{z}_{h,t})$, where the decoder $\mathcal{D}$ is equivariant and the property predictor $\mathcal{A}$ is invariant. Defining $a_{x,t}, a_{h,t} = \nabla \log p_t(y|\mathbf{z}_{x,t}, \mathbf{z}_{h,t})$ and $a'_{x,t}, a'_{h,t} = \nabla \log p_t(y|\mathbf{R}\mathbf{z}_{x,t}, \mathbf{z}_{h,t})$, then the conditional score function is orthogonal equiavariant such that $\mathbf{R}a_{x,t}, a_{h,t} = a'_{x,t}, a'_{h,t}$.*

Then, building upon the conclusion drawn by previous work (Bao et al., 2022), we can also demonstrate that our SDE is an equivariant guided SDE.

## 4.2 MULTI-CONDITION GUIDANCE

We start to analyze the multiple-property situation depicted in Fig. 3(a), which involves two desired properties, $y_1$ and $y_2$, with $y_2$ being dependent on $y_1$. We aim to compute

$$\nabla \log p_t(\mathbf{z}_t|y_1, y_2) = \nabla \log p_t(\mathbf{z}_t) + \nabla \log p_t(y_1, y_2|\mathbf{z}_t). \tag{9}$$

when we only have the time-independent property functions for clean data: $p(y_1|\mathbf{z}_0) = \ell_1(\mathbf{z}_0, y_1)$ and $p(y_2|\mathbf{z}_0) = \ell_2(\mathbf{z}_0, y_2)$. Thus, we need to factorize $p_t(y_1, y_2|\mathbf{z}_t)$ to exploit property functions:

$$
\begin{aligned}
p_t(y_1, y_2|\mathbf{z}_t) &= \int_{\mathbf{z}_0} p(y_1, y_2, \mathbf{z}_0|\mathbf{z}_t)\mathrm{d}\mathbf{z}_0 = \int_{\mathbf{z}_0} p(y_1, y_2|\mathbf{z}_0, \mathbf{z}_t)p(\mathbf{z}_0|\mathbf{z}_t)\mathrm{d}\mathbf{z}_0 \\
&= \int_{\mathbf{z}_0} p(y_1, y_2|\mathbf{z}_0)p(\mathbf{z}_0|\mathbf{z}_t)\mathrm{d}\mathbf{z}_0 = \int_{\mathbf{z}_0} p(y_1|\mathbf{z}_0)p(y_2|\mathbf{z}_0, y_1)p(\mathbf{z}_0|\mathbf{z}_t)\mathrm{d}\mathbf{z}_0 \\
&\approx p(y_1|\hat{\mathbf{z}}_0)p(y_2|\hat{\mathbf{z}}_0, y_1)
\end{aligned}
\tag{10}
$$

where $\hat{\mathbf{z}}_0 = \mathbb{E}_{\mathbf{z}_0 \sim p(\mathbf{z}_0|\mathbf{z}_t)}[\mathbf{z}_0]$. We employ the property function $\ell_1(\hat{\mathbf{z}}_0, y_1)$ to compute $p(y_1|\hat{\mathbf{z}}_0)$, which is the same as the single objective. But we do not have a property function $\ell_2((\hat{\mathbf{z}}_0, y_1), y_2)$ that can input $\mathbf{z}_0$ and $y_1$ simultaneously.

To address this challenge, we try to combine the information in $(\hat{\mathbf{z}}_0, y_1)$ to remap a new distribution of samples $\mathbf{z}_0'$, in order to transfer $p_t(y_2|\hat{\mathbf{z}}_0, y_1)$ into $p_t(y_2|\mathbf{z}_0')$. Specifically, we compute the expectation of $\mathbf{z}_0'$ with the following proposition and the proof is provided in Appendix A.2.1:

**Proposition 2.** *Suppose the prior distribution $p(\hat{\mathbf{z}}_0|\mathbf{z}_0', y_1) \sim \mathcal{N}(\mathbf{z}_0', r_t^2 I)$ and first-order expansion of the loss function $p(y_1|\mathbf{z}_0') = \exp(-\ell_1(\hat{\mathbf{z}}_0, y_1) - (\mathbf{z}_0' - \hat{\mathbf{z}}_0)^T \nabla \ell_1(\hat{\mathbf{z}}_0, y_1))$, we have*

$$\mathbb{E}_{\mathbf{z}_0 \sim p(\mathbf{z}_0|\hat{\mathbf{z}}_0, y_1)}[z_0] = \hat{\mathbf{z}}_0 - r_t^2 \nabla \ell_1(\hat{\mathbf{z}}_0, y_1)$$

Table 1: The mean absolute error (MAE) in single-objective 3D molecule generation tasks

| Method | MAE↓ | Method | MAE↓ | Method | MAE↓ |
|---|---|---|---|---|---|
| $C_v$ ($\frac{\text{cal}}{\text{mol}}$K) | | $\mu$ (D) | | $\alpha$ (Bohr$^3$) | |
| U-bound | 6.879±0.015 | U-bound | 1.613±0.003 | U-bound | 8.98±0.02 |
| #Atoms | 1.971 | #Atoms | 1.053 | #Atoms | 3.86 |
| Cond. EDM | 1.065±0.010 | Cond. EDM | 1.123±0.013 | Cond. EDM | 2.78±0.04 |
| Cond. GeoLDM | 1.025 | Cond. GeoLDM | 1.108 | Cond. GeoLDM | 2.37 |
| EEGSDE ($s$=10) | 0.941±0.005 | EEGSDE ($s$=2) | 0.777±0.007 | EEGSDE ($s$=3) | 2.50±0.02 |
| MuDM | **0.290**±0.024 | MuDM | **0.333**±0.015 | MuDM | **0.43**±0.07 |
| L-bound | 0.040 | L-bound | 0.043 | L-bound | 0.09 |
| $\Delta\varepsilon$ (meV) | | $\varepsilon_{\text{HOMO}}$ (meV) | | $\varepsilon_{\text{LUMO}}$ (meV) | |
| U-bound | 1464±4 | U-bound | 645±41 | U-bound | 1457±5 |
| #Atoms | 866 | #Atoms | 426 | #Atoms | 813 |
| Cond. EDM | 671±5 | Cond. EDM | 371±2 | Cond. EDM | 601±7 |
| Cond. GeoLDM | 587 | Cond. GeoLDM | 340 | Cond. GeoLDM | 522 |
| EEGSDE ($s$=3) | 487±3 | EEGSDE ($s$=1) | 302±2 | EEGSDE ($s$=3) | 447±6 |
| MuDM | **85**±6 | MuDM | **72**±4 | MuDM | **133**±11 |
| L-bound | 65 | L-bound | 39 | L-bound | 36 |

$\alpha = 60.9$  $\alpha = 68.3$  $\alpha = 72.3$  $\alpha = 77.1$  $\alpha = 83.3$  $\alpha = 90.3$  $\alpha = 96.7$

Figure 4: Generated molecules conditioned on the single objective (polarizability $\alpha$)

In summary, we arrive at the approximation $\nabla \log p_t(y_1, y_2|\mathbf{z}_t) \approx \nabla \log p(y_1|\hat{\mathbf{z}}_0) + \nabla \log p(y_2|\mathbf{z}'_0)$, where $\hat{\mathbf{z}}_0 = \frac{1}{\sqrt{\bar{\alpha}_t}}(\mathbf{z}_t + (1 - \bar{\alpha}_t)\nabla_{\mathbf{z}_t} \log p_t(\mathbf{z}_t))$ and $\mathbf{z}'_0 = \hat{\mathbf{z}}_0 - r_t^2 \nabla \ell_1(\hat{\mathbf{z}}_0, y_1)$.

The same procedure can be extended to analyze more than two properties or other types of property relations, when the probabilistic graph of properties is determined:

In Fig. 3(b), we have $p(y_1, y_2|\mathbf{z}_t) \approx p(y_1|\hat{\mathbf{z}}_0)p(y_2|\hat{\mathbf{z}}_0)$ when $y_1$ and $y_2$ are independent. Thus, we can simply use the gradients of the linear combination of the two property functions to guide.

In Fig. 3(c), we consider three properties $y_1, y_2$ and $y_3$, with $y_3$ being dependent on $y_1$ and $y_2$. We express $p(y_1, y_2, y_3|\mathbf{z}_t) \approx p(y_1|\hat{\mathbf{z}}_0)p(y_2|\hat{\mathbf{z}}_0)p(y_3|\hat{\mathbf{z}}_0, y_1, y_2)$. Here $p(y_3|\hat{\mathbf{z}}_0, y_1, y_2)$ is approximated by Proposition 2, and we compute $\mathbf{z}'_0(y_3) = \hat{\mathbf{z}}_0 - r_t^2[\nabla \ell_1(\hat{\mathbf{z}}_0, y_1) + \nabla \ell_2(\hat{\mathbf{z}}_0, y_2)]$ for $y_3$. Subsequently, the final conditional score function is computed by $\nabla \ell_1(\hat{\mathbf{z}}_0, y_1) + \nabla \ell_2(\hat{\mathbf{z}}_0, y_2) + \nabla \ell_3(\mathbf{z}'_0(y_3), y_3)$.

In Fig. 3(d), we also have three properties $y_1, y_2$ and $y_3$, with $y_3$ being dependent on $y_2$, and $y_2$ being dependent on $y_1$. The term $p(y_1, y_2, y_3|\mathbf{z}_t)$ is approximated by $p(y_1|\hat{\mathbf{z}}_0)p(y_2|\hat{\mathbf{z}}_0, y_1)p(y_3|\hat{\mathbf{z}}_0, y_2)$. We can apply Proposition 2 to calculate $\mathbf{z}'_0(y_2)$ and $\mathbf{z}'_0(y_3)$ separately, where $\mathbf{z}'_0(y_2) = \hat{\mathbf{z}}_0 - r_t^2 \nabla \ell_1(\hat{\mathbf{z}}_0, y_1)$, and $\mathbf{z}'_0(y_3) = \hat{\mathbf{z}}_0 - r_t^2 \nabla \ell_2(\hat{\mathbf{z}}_0, y_2)$. The final conditional score function is computed by $\nabla \ell_1(\hat{\mathbf{z}}_0, y_1) + \nabla \ell_2(\mathbf{z}'_0(y_2), y_2) + \nabla \ell_3(\mathbf{z}'_0(y_3), y_3)$.

Currently, we can handle any direct acyclic probabilistic graphs. In addition, MC sampling is used to improve multi-condition guidance. By sampling multiple $\hat{\mathbf{z}}_0$, we obtain a more accurate estimation of the gradients for each property. Moreover, we also adapt the weighted sum of gradients in practice. These weights are hyperparameters and remain fixed during the generative process.

## 5 EXPERIMENTS

In this section, we conduct comprehensive experiments on single and multiple-conditioned molecule generation tasks to evaluate our proposed method MUDM. The pseudo-code and hyperparameters are provided in Appendix A.1 and A.3, and the code will be published later.

Table 2: The mean absolute error (MAE) in multi-objective 3D molecule generation tasks.

| Multi-objective Tasks | | | Metrics | Baselines | | |
|---|---|---|---|---|---|---|
| Property 1 | Property 2 | Correlation | MAE ↓ | Conditional EDM | EEGSDE | MuDM |
| $C_v$ ($\frac{\text{cal}}{\text{mol}}$K) | $\mu$ (D) | 0.42 | MAE 1 | 1.079 | **0.981** | 1.466 |
| | | | MAE 2 | 1.156 | 0.912 | **0.687** |
| $\Delta\varepsilon$ (meV) | $\mu$ (D) | -0.34 | MAE 1 | 683 | 563 | **554** |
| | | | MAE 2 | 1.130 | 0.866 | **0.578** |
| $\alpha$ (Bohr$^3$) | $\mu$ (D) | -0.24 | MAE 1 | 2.760 | 2.610 | **1.326** |
| | | | MAE 2 | 1.158 | 0.855 | **0.519** |
| $\varepsilon_{\text{HOMO}}$ (meV) | $\varepsilon_{\text{LUMO}}$ (meV) | 0.22 | MAE 1 | 372 | 335 | **317** |
| | | | MAE 2 | 594 | 517 | **455** |
| $\varepsilon_{\text{LUMO}}$ (meV) | $\mu$ (D) | -0.40 | MAE 1 | 610 | **526** | 575 |
| | | | MAE 2 | 1.143 | 0.860 | **0.497** |
| $\varepsilon_{\text{LUMO}}$ (meV) | $\Delta\varepsilon$ (meV) | 0.89 | MAE 1 | 1097 | 546 | **361** |
| | | | MAE 2 | 712 | 589 | **228** |
| $\varepsilon_{\text{HOMO}}$ (meV) | $\Delta\varepsilon$ (meV) | -0.24 | MAE 1 | 578 | 567 | **262** |
| | | | MAE 2 | 655 | **323** | 489 |

## 5.1 SETUP

**Dataset:** We perform conditional molecule generation on QM9 (Ramakrishnan et al., 2014), a dataset of over 130K molecules and 6 corresponding quantum properties. Following previous research, we split the dataset into training, valid, and test sets, each including 100K, 18K, and 13K samples respectively. The training set is further separated into 2 equal halves, $D_a$ and $D_b$, to avoid information leak in the training phase. The half $D_a$ is used to train the ground-truth property prediction network while $D_b$ is for the training of the diffusion model.

**Metrics:** We use the mean absolute error (MAE) to evaluate the difference between the given condition and the property of generated molecules.

**Baselines:** In this study, we compare MᴜDM with several competitive previous works. The conditional EDM trains the diffusion model on $D_b$, given the molecule and its property ($[x, h], c$). In this case, the conditional EDM needs to train separate diffusion models for each property. Similarly, the conditional GEOLDM is also required to train six diffusion models. EEGSDE trains one diffusion model and six time-dependent property functions on $D_b$ for each property to guide the sampling phase. Occasionally, EEGSDE fine-tunes the diffusion model based on property functions to enhance performance. Moreover, our proposed method MᴜDM trains one diffusion model and six time-independent property functions on $D_b$ without the need for fine-tuning.

Additionally, EDM reported "#Atoms", as well as the upper-bound and lower-bound as baselines. The "L-bound" baseline refers to the MAE of the ground-truth property function on $D_b$, while the "U-bound" baseline shuffles the labels and calculates the MAE on $D_b$. The "#Atoms" baseline predicts the property based on the number of atoms in a molecule.

## 5.2 SINGLE-CONDITIONED MOLECULE GENERATION

We examine six quantum properties in QM9 as our single objective: heat capacity $C_v$, dipole moment $\mu$, polarizability $\alpha$, highest occupied molecular orbital energy $\varepsilon_{\text{HOMO}}$, lowest unoccupied molecular orbital energy $\varepsilon_{\text{LUMO}}$, and HOMO-LUMO gap $\Delta\varepsilon$. As shown in Table 1, our method MᴜDM achieved the best performance on each quantum property compared to conditional EDM, GeoLDM and EEGSDE. This result highlights the potential of using time-independent property functions to directly guide the generative process. Additionally, we also confirm the effectiveness of both DPS and MC sampling in providing accurate guidance for property prediction.

In terms of efficiency, training a diffusion model for a new property using conditional EDM and GeoLDM takes $\sim 2$ days on a single A100 GPU. Meanwhile, obtaining a specialized time-dependent property function using EEGSDE also takes several days. In contrast, our method MᴜDM directly

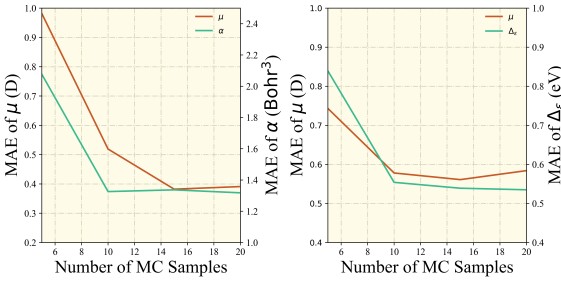
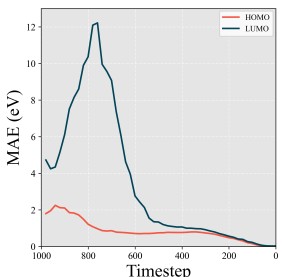

Figure 5: Impact of the number of MC samples

Figure 6: MAE during the generative process

utilized the off-the-shelf property function without requiring extra training. This approach allows for faster predictions, providing researchers in the area of drug discovery with a quick and flexible way to search for new molecules with desired properties.

Finally, we present a visualization of generated molecules with varying $\alpha$ as the condition in Fig. 4. The polarizability $\alpha$ is defined as the ratio of its induced dipole moment to the local electric field. Generally, molecules with less isometric shapes have larger $\alpha$ values. We set $\alpha \in [60, 100]$ to observe that the generated molecules become less isometric, which is consistent with the expectation.

### 5.3 MULTIPLE-CONDITIONED MOLECULE GENERATION

We investigate seven combinations of quantum properties. We first compute the correlation between each pair on $D_b$. We discovered that all of these combinations were correlated, so we used the situation depicted in Fig. 3(a) to model the variables $y_1$ and $y_2$, with $y_2$ being dependent on $y_1$. Our proposed method MUDM outperformed conditional EDM and EEGSDE in most cases, which demonstrates the effectiveness of the proposed multiple-objective guidance.

As for efficiency, our multi-objective guidance essentially does not increase the computation time and cost compared to single-objective tasks. In contrast, conditional EDM and EEGSDE require retraining the diffusion model for each combination, which needs a huge computation resource.

Two visualizations of generated molecules with multiple objectives are shown in Appendix A.7.

### 5.4 ABLATION STUDY

We present two ablation studies in this section and provide more ablation studies in Appendix A.4. The first one is the impact of the number of Monte Carlo (MC) samples. Fig. 5 shows the performance of multi-objective tasks with different #samples. With the increased #samples, the performance becomes better. However, this improvement came at the cost of increased time consumption. Analyzing the trend in Fig. 5, we found that #samples = 10 is the balanced choice between performance and time efficiency.

We also observed the difference $|y - \mathcal{A}(\mathcal{D}(\hat{\mathbf{z}}_0(\mathbf{z}_t)))|$ during the generative process in Fig. 6. We found that the difference initially dramatically increases in $[1000, 500]$ and then subsequently decreases gradually in $[500, 0]$. This observation aligns with the finding presented in Fig. 2, which indicates the presence of two distinct stages. Consequently, calculating property values in the chaotic stage yields inaccurate results. We should only provide guidance during the semantic stage.

## 6 CONCLUSION

In this paper, we proposed MUDM, a more flexible and efficient approach for generating 3D molecules with desired properties. MUDM is capable of handling both single and multiple property objectives for molecule generation. The experiments further validate the effectiveness of MUDM. As a future direction, MUDM can be applied to more realistic properties, such as chirality and binding affinity with protein targets, expanding its potential applications in drug and material discovery.

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

## A APPENDIX

### A.1 PSEUDO-CODE

The algorithm below illustrates the sampling process of MuDM conditioned on the single property, and two properties.

---

**Algorithm 1** Sampling Algorithm of MuDM for single condition

---

1: **Input:** decoder network $\mathcal{D}_\xi$, denoising network $\epsilon_\theta$, condition $y$, property predictor $\mathcal{A}$
2: $\mathbf{z}_{\mathrm{x},T}, \mathbf{z}_{\mathrm{h},T} \sim \mathcal{N}(\mathbf{0}, \boldsymbol{I})$
3: **for** $t$ in $T, T-1, \cdots, 1$ **do**
4:    $\epsilon \sim \mathcal{N}(\mathbf{0}, \boldsymbol{I})$         {Single Condition Guided Latent Denoising Loop}
5:    Subtract center of gravity from $\epsilon_\mathrm{x}$ in $\epsilon = [\epsilon_\mathrm{x}, \epsilon_\mathrm{h}]$
6:    $\mathbf{z}'_{t-1} = \frac{1}{\sqrt{1-\beta_t}}(\mathbf{z}_t - \frac{\beta_t}{\sqrt{1-\alpha_t^2}}\epsilon_\theta(\mathbf{z}_t, t)) + \rho_t\epsilon$
7:    $\hat{\mathbf{z}}_0 = \frac{1}{\alpha_t}(\mathbf{z}_t + \sqrt{1-\alpha_t^2}\epsilon_\theta(\mathbf{z}_t, t))$
8:    $\mathbf{z}_{t-1} = \mathbf{z}'_{t-1} + \nabla_{\mathbf{z}_t}\log(\frac{1}{n}\sum_{i=1}^n\exp(-\ell(\mathcal{A}(\mathbf{z}_0^i), y)), \mathbf{z}_0^i \sim \mathcal{N}(\hat{\mathbf{z}}_0, r_t^2\mathbf{I})$
9: **end for**
10: $\mathbf{x}, \mathbf{h} \sim p_\xi(\mathbf{x}, \mathbf{h}|\mathbf{z}_{\mathrm{x},0}, \mathbf{z}_{\mathrm{h},0})$         {Decoding}
11: **return** $\mathbf{x}, \mathbf{h}$

---

---

**Algorithm 2** Sampling Algorithm of MuDM for two conditions

---

1: **Input:** decoder network $\mathcal{D}_\xi$, denoising network $\epsilon_\theta$, first condition $y_1$, first property predictor $\mathcal{A}_1$, second condition $y_2$, second property predictor $\mathcal{A}_2$
2: $\mathbf{z}_{\mathrm{x},T}, \mathbf{z}_{\mathrm{h},T} \sim \mathcal{N}(\mathbf{0}, \boldsymbol{I})$
3: **for** $t$ in $T, T-1, \cdots, 1$ **do**
4:    $\epsilon \sim \mathcal{N}(\mathbf{0}, \boldsymbol{I})$         {Two Conditions Guided Latent Denoising Loop}
5:    Subtract center of gravity from $\epsilon_\mathrm{x}$ in $\epsilon = [\epsilon_\mathrm{x}, \epsilon_\mathrm{h}]$
6:    $\mathbf{z}'_{t-1} = \frac{1}{\sqrt{1-\beta_t}}(\mathbf{z}_t - \frac{\beta_t}{\sqrt{1-\alpha_t^2}}\epsilon_\theta(\mathbf{z}_t, t)) + \rho_t\epsilon$
7:    $\hat{\mathbf{z}}_0 = \frac{1}{\alpha_t}(\mathbf{z}_t + \sqrt{1-\alpha_t^2}\epsilon_\theta(\mathbf{z}_t, t))$
8:    $\nabla\ell_1(\hat{\mathbf{z}}_0, y_1) = \nabla_{\mathbf{z}_t}\log(\frac{1}{n}\sum_{i=1}^n\exp(-\ell(\mathcal{A}_1(\mathbf{z}_0^i), y_1)), \mathbf{z}_0^i \sim \mathcal{N}(\hat{\mathbf{z}}_0, r_t^2\mathbf{I})$
9:    $\hat{\mathbf{z}}'_0 = \hat{\mathbf{z}}_0 + \frac{1-\alpha_t^2}{\alpha_t}\nabla\ell_1(\hat{\mathbf{z}}_0, y_1)$
10:   $\nabla\ell_2(\hat{\mathbf{z}}'_0, y_2) = \nabla_{\mathbf{z}_t}\log(\frac{1}{n}\sum_{i=1}^n\exp(-\ell(\mathcal{A}_2(\mathbf{z}_0^i), y_2)), \mathbf{z}_0^i \sim \mathcal{N}(\hat{\mathbf{z}}'_0, r_t^2\mathbf{I})$
11:   $\mathbf{z}_{t-1} = \mathbf{z}'_{t-1} + \nabla\ell_1(\hat{\mathbf{z}}_0, y_1) + \nabla\ell_2(\hat{\mathbf{z}}'_0, y_2)$
12: **end for**
13: $\mathbf{x}, \mathbf{h} \sim p_\xi(\mathbf{x}, \mathbf{h}|\mathbf{z}_{\mathrm{x},0}, \mathbf{z}_{\mathrm{h},0})$         {Decoding}
14: **return** $\mathbf{x}, \mathbf{h}$

---

### A.2 PROOFS

#### A.2.1 PROOF OF PROPOSITION 2

**Proposition 3.** *Suppose the prior distribution $p(\hat{\mathbf{z}}_0|\mathbf{z}'_0, y_1) \sim \mathcal{N}(\mathbf{z}'_0, r_t^2 I)$ and first-order expansion of the loss function $p(y_1|\mathbf{z}'_0) = \exp(\ell_1(\hat{\mathbf{z}}_0, y_1) + (\mathbf{z}'_0 - \hat{\mathbf{z}}_0)^T\nabla\ell_1(\hat{\mathbf{z}}_0, y_1))$, we have*

$$\mathbb{E}_{\mathbf{z}_0 \sim p(\mathbf{z}_0|\hat{\mathbf{z}}_0, y_1)}[z_0] = \hat{\mathbf{z}}_0 - r_t^2\nabla\ell_1(\hat{\mathbf{z}}_0, y_1)$$

*Proof.* We have that

$$p(\mathbf{z}'_0|\hat{\mathbf{z}}_0, y_1) = \frac{p(\hat{\mathbf{z}}_0|\mathbf{z}'_0, y_1)p(\mathbf{z}'_0|y_1)p(y_1)}{p(y_1, \hat{\mathbf{z}}_0)}$$

where both $p(y_1)$ and $p(y_1, \hat{\mathbf{z}}_0)$ are normalizing constants. Then we have

$$
\begin{aligned}
p(\mathbf{z}_0' = \mathbf{k}|\hat{\mathbf{z}}_0, y_1) &\propto p(\hat{\mathbf{z}}_0|\mathbf{z}_0' = \mathbf{k}, y_1)p(y_1|\mathbf{z}_0' = \mathbf{k})p(\mathbf{z}_0' = \mathbf{k}) \\
&\propto \exp\left(-\frac{(\mathbf{k} - \hat{\mathbf{z}}_0)^2}{r_t^2}\right) \cdot \exp(-\ell_1(\hat{\mathbf{z}}_0, y_1) - (\mathbf{k} - \hat{\mathbf{z}}_0)^T \nabla \ell_1(\hat{\mathbf{z}}_0, y_1)) \\
&\propto \exp\left(-\frac{(\mathbf{k} - \hat{\mathbf{z}}_0)^2}{r_t^2} - (\mathbf{k} - \hat{\mathbf{z}}_0)^T \nabla \ell_1(\hat{\mathbf{z}}_0, y_1)\right).
\end{aligned}
$$

As the term within exp is a quadratic function, the expectation is achieved at the maximum point, i.e.,

$$
\mathbb{E}_{\mathbf{z}_0 \sim p(\mathbf{z}_0|\hat{\mathbf{z}}_0, y_1)}[z_0] = \underset{\mathbf{k}}{\operatorname{argmax}} - \frac{(\mathbf{k} - \hat{\mathbf{z}}_0)^2}{r_t^2} - (\mathbf{k} - \hat{\mathbf{z}}_0)^T \nabla \ell_1(\hat{\mathbf{z}}_0, y_1) = \hat{\mathbf{z}}_0 - r_t^2 \nabla \ell_1(\hat{\mathbf{z}}_0, y_1)
$$

$\square$

### A.2.2 PROOF OF PROPOSITION 1

**Proposition 4.** *Suppose the loss function of the property $f = \ell(\mathcal{A}(\mathcal{D}(\cdot)), y)$ is invariant such that $f(\mathbf{R}\mathbf{z}_{x,t}, \mathbf{z}_{h,t}) = f(\mathbf{z}_{x,t}, \mathbf{z}_{h,t})$, where the decoder $\mathcal{D}$ is equivariant and the property predictor $\mathcal{A}$ is invariant. Defining $a_{x,t}, a_{h,t} = \nabla \log p_t(y|\mathbf{z}_{x,t}, \mathbf{z}_{h,t})$ and $a_{x,t}', a_{h,t}' = \nabla \log p_t(y|\mathbf{R}\mathbf{z}_{x,t}, \mathbf{z}_{h,t})$, then the conditional score function is orthogonal equiavariant such that $\mathbf{R}a_{x,t}, a_{h,t} = a_{x,t}', a_{h,t}'$.*

*Proof.* For the decoder $\mathcal{D}$, we have $\mathbf{R}x, h = \mathcal{D}(\mathbf{R}z_x, z_h)$. When the loss function of the property $f = \ell(\mathcal{A}(\cdot), y)$ is invariant, we have $f(\mathbf{R}x, h, y) = f(x, h, y)$. We take the gradient w.r.t $x$ to both sides and obtain

$$
\begin{aligned}
\nabla_x f(x, h, y) &= \nabla_x x' \cdot \nabla_{x'} f(x', h, y) \quad \text{where } x' = \mathbf{R}x, \\
\nabla_x f(x, h, y) &= \mathbf{R}^{\mathrm{T}} \nabla_{x'} f(x', h, y) \quad \text{where } x' = \mathbf{R}x.
\end{aligned}
$$

Multiplying $\mathbf{R}$ to both sides where $\mathbf{R}\mathbf{R}^{\mathrm{T}} = 1$, we get

$$
\mathbf{R}\nabla_x f(x, h, y) = \nabla_{x'} f(x', h, y)|_{x'=\mathbf{R}x}.
$$

Thus, $\nabla_x f(\cdot)$ is equivariant to $\mathbf{R}$. The proposition is proved. $\square$

### A.3 DETAILS FOR EXPERIMENTAL SETUPS

We didn't train any new model for the 3D molecule generation task. The pre-trained model was directly from GEOLDM and we kept the same hyperparameter settings without any additional fine-tuning. For any property guidance, we used the same diffusion model $\epsilon_\theta(\mathcal{G}_t, t)$, with the corresponding property predictor $\mathcal{A}(.) : \mathcal{G} \to \mathbb{R}$. This property predictor is not time-dependent.

We found it is important to set reasonable guidance weights for multi-condition guidance. We computed the gradients as follows:

$$
\nabla \log p_t(y_1, y_2|\mathbf{z}_t) = w_1 \nabla \log p(y_1|\hat{\mathbf{z}}_0) + w_2 \nabla \log p(y_2|\mathbf{z}_0')
$$

We listed the main sampling settings in Table 3. $r$ is the variance level used for mc sampling $\mathcal{N}(\hat{\mathbf{z}}_0, r_t^2 \mathbf{I})$. We fixed it during the sampling process.

Table 3: Hyperparameters for two conditions sampling

| Conditioned properties | Guide from | $w_1$ | $w_2$ | $r$ | MC sample size |
|---|---|---|---|---|---|
| $C_v$ ($\frac{\text{cal}}{\text{mol}}$K), $\mu$ (D) | 400 | 3 | 1 | 0.6 | 10 |
| $\Delta\varepsilon$ (meV), $\mu$ (D) | 400 | 1 | 1 | 0.3 | 10 |
| $\alpha$ (Bohr$^3$), $\mu$ (D) | 400 | 2 | 1 | 0.3 | 10 |
| $\varepsilon_{\text{HOMO}}$ (meV), $\varepsilon_{\text{LUMO}}$ (meV) | 400 | 1 | 1 | 0.3 | 10 |
| $\varepsilon_{\text{LUMO}}$ (meV), $\mu$ (D) | 400 | 1 | 1 | 0.3 | 10 |

### A.4 OTHER ABLATION STUDIES

Comprehensive ablation studies are included in this section to demonstrate the effectiveness of each design. We kept all settings the same as the main paper and only changed one setting to check the performance change.

#### A.4.1 INFLUENCE OF GUIDANCE STEP

Table 4 shows how the guidance steps influence the task with two conditions. The result indicates that there is no significant change when we guided the diffusion model from 400 steps or 1000 steps. This is because, during the chaotic stage, the property predictor cannot provide accurate predictions and thus can not provide useful information for guidance. It proves that starting from 400 steps is efficient for the guidance process and can help save inference time.

Table 4: Influence of guidance steps

| Conditioned properties | Guide from | MAE 1 | MAE 2 |
|---|---|---|---|
| $\varepsilon_{\text{HOMO}}$ (meV), $\Delta\varepsilon$ (meV) | 400 | 262 | 489 |
| | 1000 | 271 | 465 |

#### A.4.2 COMPARISON OF DPS AND MC SAMPLING

We show the result for two different sampling methods DPS and MC sampling in Table 5. It indicates the MC sampling method has a better performance. The previous work Song et al. (2023) proved that MC sampling method is a more closed estimation for $\mathbb{E}_{\mathbf{z}_0 \sim p(\mathbf{z}_0|\mathbf{z}_t)}[p(y|\mathcal{G} = \mathcal{D}(\mathbf{z}_0)]$.

Table 5: Influence of MC sampling

| Conditioned properties | Sampling method | MAE 1 | MAE 2 |
|---|---|---|---|
| $\varepsilon_{\text{LUMO}}$ (meV), $\Delta\varepsilon$ (meV) | DPS | 472 | 396 |
| | MC sampling | 361 | 228 |

#### A.4.3 INFLUENCE OF CONDITION WEIGHTS

We tested how different weight settings influence the performance in Table 6. It shows that different weights have impacts on the final results. But in most cases, we can simply set all weights as 1.

Table 6: Influence of condition weights

| Conditioned properties | $w_1$ | $w_2$ | MAE 1 | MAE 2 |
|---|---|---|---|---|
| $\varepsilon_{\text{LUMO}}$ (meV), $\Delta\varepsilon$ (meV) | 1 | 1 | 361 | 228 |
| | 1 | 2 | 540 | 456 |
| | 2 | 1 | 393 | 342 |
| | 1 | 3 | 687 | 545 |
| | 3 | 1 | 613 | 526 |

#### A.4.4 INDEPENDENT SAMPLING FOR MULTIPLE-CONDITIONED MOLECULE GENERATION

Finally, we ignore the dependency between properties and believe they are independent. Thus, the multiple-objective guidance is just a linear combination of gradients of each property. The results are shown in Table 7. We found that the performance of all the properties dramatically declined.

It verifies the effectiveness of our proposed multiple-objective guidance, which is calculated as the weighted sum of the gradient of each property with respect to the latent variable, taking into account the property dependency.

Table 7: Results of a linear combination of multi-objective guidance

| Property 1 | MAE 1 | Desc. | Property 2 | MAE 2 | Desc. |
|---|---|---|---|---|---|
| $C_v$ ($\frac{\text{cal}}{\text{mol}}$K) | 1.461 | 0.3% | $\mu$ (D) | 0.942 | -37.1% |
| $\Delta_\varepsilon$ (eV) | 0.851 | -53.6% | $\mu$ (D) | 0.591 | -2.2% |
| $\alpha$ (Bohr$^3$) | 3.012 | -127.1% | $\mu$ (D) | 0.952 | -83.4% |
| $\varepsilon_{\text{HOMO}}$ (eV) | 0.446 | -40.7% | $\varepsilon_{\text{LUMO}}$ (eV) | 0.522 | -14.7% |
| $\varepsilon_{\text{LUMO}}$ (eV) | 0.755 | -31.3% | $\mu$ (D) | 0.930 | -87.1% |

## A.5 Three-conditioned molecule generation

In our extended analysis, we introduced an additional experiment to evaluate the performance of MUDM in scenarios involving three properties, as illustrated in Table 8 and reflected in Figure 3 (c). This experiment was designed to compare scenarios with potential conflicts among properties against those without such conflicts. In the first scenario (denoted by ✗), the properties $\varepsilon_{\text{HOMO}}$, $\varepsilon_{\text{LUMO}}$, and $\Delta\varepsilon = \varepsilon_{\text{LUMO}} - \varepsilon_{\text{HOMO}}$ were independently sampled, introducing the possibility of conflicting requirements. Consequently, no molecule could perfectly satisfy the targeted properties, yet MUDM still demonstrated commendable performance. In contrast, the second scenario (denoted by ✓) did not present any inherent conflict among the three properties since $\Delta\varepsilon$ is computed from the other two properties. As expected, the performance in this conflict-free setting was superior, as indicated by the lower mean absolute errors (MAEs). These results not only underscore the robustness of MUDM in handling multi-objective tasks with complex inter-property relationships but also highlight its capability to generate feasible molecular structures even in the presence of potential property conflicts.

Table 8: The mean absolute error (MAE) for three objective 3D molecule generation tasks

| Conflict | Property 1 | Property 2 | Property 3 | MAE 1 | MAE 2 | MAE 3 |
|---|---|---|---|---|---|---|
| ✓ | $\varepsilon_{\text{HOMO}}$(meV) | $\varepsilon_{\text{LUMO}}$(meV) | $\Delta\varepsilon$ (meV) | 479 | 545 | 678 |
| ✗ | $\varepsilon_{\text{HOMO}}$(meV) | $\varepsilon_{\text{LUMO}}$(meV) | $\Delta\varepsilon$ (meV) | 283 | 392 | 255 |

## A.6 Description of the QM9 Dataset

The QM9 dataset includes 134k stable small molecules, consisting of atoms C, H, O, N and F. The statistic of the number of atoms is provided in Table 9. We can obtain atom coordinates and corresponding quantum properties for each molecule. These quantum properties are calculated at the B3LYP/6-31G(2df,p) level of quantum chemistry. Previously, QM9 was a benchmarking dataset to evaluate whether the machine learning-based method can predict accurately the quantum properties based on the molecule structure. Now it is also employed in molecular generation tasks.

Table 9: Statistic of QM9 Dataset

| | Mean | STD | Maximum |
|---|---|---|---|
| Number of atoms | 18.0 | 3.0 | 29 |
| Number of heavy atoms | 8.8 | 0.51 | 9 |

## A.7 Visualization of generated molecules with multiple objectives

We provide two visualizations of generated molecules with multiple objectives in Fig. 7. The dipole moment $\mu$ is a measure of the separation of positive and negative electrical charges, so the molecules

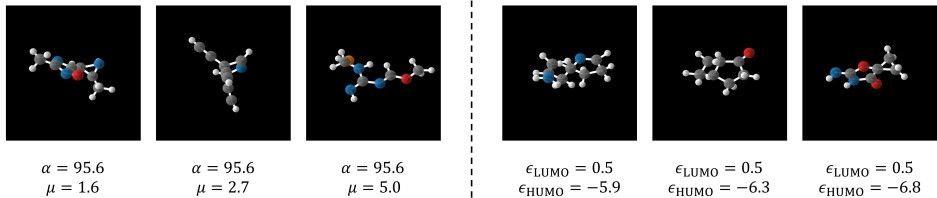

$$\begin{array}{ccc|ccc}
\alpha = 95.6 & \alpha = 95.6 & \alpha = 95.6 & \epsilon_{\text{LUMO}} = 0.5 & \epsilon_{\text{LUMO}} = 0.5 & \epsilon_{\text{LUMO}} = 0.5 \\
\mu = 1.6 & \mu = 2.7 & \mu = 5.0 & \epsilon_{\text{HUMO}} = -5.9 & \epsilon_{\text{HUMO}} = -6.3 & \epsilon_{\text{HUMO}} = -6.8
\end{array}$$

Figure 7: Generated molecules conditioned on the multiple objectives

with high dipole moment and polarizability are also asymmetrical. Besides, molecules with a high energy gap between HOMO and LUMO are generally less reactive and more stable, indicating that their constituent atoms are closely connected. These trends are observed in Fig. 7.

## A.8 RESULTS ON NOVELTY AND ATOM STABILITY

Table 10: Other Properties

| Conditioned properties | Atom stability | Novelty |
|---|---|---|
| $C_v$ ($\frac{\text{cal}}{\text{mol}}$K), $\mu$ (D) | 0.72 | 0.84 |
| $\Delta\varepsilon$ (meV), $\mu$ (D) | 0.66 | 0.92 |
| $\alpha$ (Bohr$^3$), $\mu$ (D) | 0.70 | 0.88 |
| $\varepsilon_{\text{HOMO}}$ (meV), $\varepsilon_{\text{LUMO}}$ (meV) | 0.67 | 0.93 |
| $\varepsilon_{\text{LUMO}}$ (meV), $\mu$ (D) | 0.68 | 0.90 |

To provide a comprehensive evaluation, we include results on the metrics of novelty and atom stability. Novelty, as defined by (Simonovsky & Komodakis, 2018), measures the percentage of generated molecules that are not present in the training dataset. Atom stability, according to (Hoogeboom et al., 2022), assesses the percentage of atoms within the molecules that exhibit correct valency. Our approach achieves a similar level of novelty compared with EEGSDE. However, we observe a detrimental effect on atom stability with the current guidance method. This issue is recognized as a limitation and should be considered in future work.

