# OpenReview forum: "Training-free Multi-objective Diffusion Model for 3D Molecule Generation"
_ICLR.cc/2024/Conference — ICLR 2024 poster_

### Official Review · Reviewer_ufcA · 2023-10-31

**Soundness:** 3 good
**Presentation:** 3 good
**Contribution:** 3 good
**Rating:** 8
**Confidence:** 3

**Summary:**

This paper proposes a method, called MuDM, for generating 3d molecules with both single and multiple desired quantum properties based on specially designed score function as guidance. For the single property optimization task, they followed previous work from the images domain to utilize the Monte-Carlo (MC) sampling to estimate the conditional score function which serves as the guidance. Besides, they devised a confidence boundary based on the prediction of the property function to divide the generation process into two stages, i.e., the chaotic stage and the semantic stage, and the guidance only works in the second stage. For optimizing multiple properties, they model the relationships between the multiple properties as probabilistic graph and factorized the conditional score function according to different probabilistic graphs. Their main contribution is extending the guidance on single molecular property to multiple properties simultaneously by building a probabilistic graph to model the relationships between various properties. They conducted experiments for both single-objective and multi-objective optimization tasks to conclude that the designed guidance outperforms other methods and is more efficient. Lastly, ablation studies were carried out to study the impact of the number of MC samples and to demonstrate the existence of the two stages in generation process and the effectiveness of the probabilistic graph.

**Strengths:**

Their main contribution is extending the guidance on single molecular property to multiple properties simultaneously by building a probabilistic graph to model the relationships between various properties. The proposed method is novel and interesting.

**Weaknesses:**

The experiments can be further improved.

**Questions:**

Certain aspects of the experiments require further clarification.

1、Can the authors provide more information about the property checker and explain how the boundary is determined.

2、In ablation studies, the reviewer agrees that the MAE curve for HOMO demonstrate the existence of the two stage in generation process. However, for the LUMO curve, no significant increase is observed, what are the possible explanations?

3、The author conducted experiments focusing on the simultaneous optimization of two interrelated properties, as illustrated in Figure 3(a). Could they extend their experiments to cover the scenarios presented in Figure 3(c) where y1 and y2 are not related?

4、In proof of proposition 4, why A(Rx, h) equals to RA(x, h) while the property predictor A is invariant? Besides, there is a typo on the numerator after the second equals sign.

Minor comments:
1、In equation 8, n indicates the number of samples, not m.
2、A typo at the beginning of the last paragraph in related works.

---

> ### Author Response · Authors · 2023-11-22
> **Response to Reviewer ufcA**
>
> First and foremost, we express our heartfelt gratitude for your warm acknowledgment of the novelty of building probabilistic graphs to model the relationships between various properties for multi-conditioned generation task. Here are our responses to your questions.
>
> > Q1: Can the authors provide more information about the property checker and explain how the boundary is determined.
>
> Thanks for your constructive suggestions! We do believe that the border decision of chaotic and semantic stages is important. To do this, we visualize generated molecules in the range of $t \in [1000,0]$ at 50-step intervals using the pre-trained diffusion model, without defining an objective. We empirically found most generated molecules in $t=400$ are already widely distributed and have certain shapes. Thus, we established 400 as the boundary. For other noise schedules, we can also first visualize the generated process, and decide an appropriate range.
>
>
> The property checker is to evaluate whether the property function of the input (i.e., the molecules generated at $t$) is out of range. If the output value falls outside the property function's range, this value will not be used to guide the direction to prevent instability of the whole process. We provided more details about the property checker on page 5 of the revised manuscript.
>
>
> The main reason that we distinguish chaotic and semantic stages, is that the judgment of the property checker always returns 'not guide' during the chaotic stage. To improve efficiency, we explicitly provide guidance only during the semantic stage. An ablation study was conducted in A.4.1  on two conditioned properties HOMO/Gap. The results indicated that there is no significant difference in guidance, whether the chaotic stage is included or not.
>
> | Guide from | MAE 1 | MAE 2 |
> |------------|-------|-------|
> | 400        | 262   | 489   |
> | 1000       | 271   | 465   |
>
>
>
> > Q2: In ablation studies, the reviewer agrees that the MAE curve for HOMO demonstrate the existence of the two stage in generation process. However, for the LUMO curve, no significant increase is observed, what are the possible explanations?
>
> Thanks for your question! The value range of LUMO in the training dataset is indeed larger than that of HOMO. In fact, the range of HOMO is small [-8,-4] compared to the LUMO range [-4,4].  This discrepancy makes the neural predictor for HOMO less sensitive to input molecules. Even when the input deviates significantly from valid molecules, the predictor doesn't produce extreme values, which results in a seemingly small MAE. However, it's worth noting that the MAE of HOMO decreases rapidly after 400 steps.
>
> > Q3: The author conducted experiments focusing on the simultaneous optimization of two interrelated properties, as illustrated in Figure 3(a). Could they extend their experiments to cover the scenarios presented in Figure 3(c) where y1 and y2 are not related?
>
> Thanks for your comments! We introduced an additional experiment for y1 = HOMO, y2=LUMO, and y3=Gap, where y1, y2, y3 satisfy the relation in Figure 3(c). We designed two scenarios, one is that y3 is conflicted with y1 and y2, and the other is y3 is consistent with y1 and y2. In the first scenario, the properties HOMO, LUMO and Gap were independently sampled, introducing the possibility of conflicting requirements. Consequently, no molecule could perfectly satisfy the targeted properties, yet MUDM still demonstrated commendable performance. In contrast, the second scenario did not present any inherent conflict among the three properties since Gap is computed from the other two properties. As expected, the performance in this conflict-free setting was superior, as indicated by the lower mean absolute errors (MAEs). These results not only underscore the robustness of MUDM in handling multi-objective tasks with complex inter-property relationships but also highlight its capability to generate feasible molecular structures even in the presence of potential property conflicts.
>
> | Conflict | MAE 1 | MAE 2 | MAE 3 |
> |----------|-------|-------|-------|
> | Yes      | 479   | 545   | 678   |
> | No       | 283   | 392   | 255   |
>
>
> > Q4: In proof of proposition 4, why A(Rx, h) equals to RA(x, h) while the property predictor A is invariant? Besides, there is a typo on the numerator after the second equals sign.
>
> Thanks for your questions! We have revised the error and provided new proof demonstrating our SDE is also equivariant. Please find it in Appendix A.2.2.
>
>
> > Q5: 1、In equation 8, n indicates the number of samples, not m. 2、A typo at the beginning of the last paragraph in related works.
>
> We really appreciate your careful reading! We have revised these typos in the manuscript.
>
> We believe your suggestions are helpful. Thanks a lot for your efforts on our submission. We revised the manuscript based on your suggestions. Your support means a lot to us.

---

### Official Review · Reviewer_7aDk · 2023-10-31

**Soundness:** 2 fair
**Presentation:** 2 fair
**Contribution:** 3 good
**Rating:** 6
**Confidence:** 4

**Summary:**

This work introduces a new property guidance mechanism for diffusion models that: (a) does not require to retrain a diffusion model, (b) uses time-independent property estimators and (c) works with arbitrary combinations of properties and is able to manage complex relationships between them. While I find this topic very important and the proposed guidance mechanism extremely useful (potentially more usable than other existing ones), I think that several important technical aspects of the proposed mechanism are not sufficiently studied (or explained in the text). Besides, in my opinion, the experimental part of this work can be improved. I am happy to reconsider the score if the points listed in the "Weaknesses" section will be addressed.

**Strengths:**

1. I find this topic very important and relevant
2. Authors provide a solid mathematical framework for the proposed guidance mechanism
3. The proposed method clearly outperforms other state-of-the-art models in single-objective 3D molecule generation

**Weaknesses:**

1. I think the question about chaotic and semantic stages is very important. Authors briefly talk about it in Section 4.1 and experimentally illustrate the importance of this issue in Figure 7. Besides, authors mention a certain "property checker" twice in Section 4.1, however never expand on it. I find it very critical to have a clear understanding of how the sampling stage is taken into account. I would be also interested in seeing the corresponding ablation study (e.g. if to take / not to take into account the chaotic stage).

2. While one of the main contributions of this work is multi-condition guidance mechanism, in my opinion a more solid experimental validation of the proposed method is required:
    * Combinations of properties considered in Table 2 look a bit strange. Why is dipole moment used in 4 out of 5 considered combinations as Property 2?
    * Besides, correlation 0.22 doesn't seem too high – aren't there better combinations?
    * Combinations $(\Delta\epsilon, \epsilon_{\text{HOMO}})$ and $(\Delta\epsilon, \epsilon_{\text{LUMO}})$ seem to be good candidates to illustrate the dependency from Figure 3(a).
    * It would be interesting to see some experiments with combination from Figure 3(c). Again, $(\Delta\epsilon, \epsilon_{\text{HOMO}}, \epsilon_{\text{LUMO}})$ seems to be a good candidate for it.

3. _"We found it is important to set reasonable guidance weights for multi-condition guidance"_  – how did authors find these weights and how much do the final results depend on these weights?

4. How does the sampling time depend on the number of MC samples? I would be also interested to see the comparison of the sampling time with other methods.

5. As authors discuss in Section 4.1, point estimation used in DPS is very inaccurate, especially in case of 3D molecule generation. It would be interesting to see some experiments with DPS (e.g. for single property conditioning) and comparison with the proposed MC-based method.

**Questions:**

1. I had an impression that adding another gradient to the denoising process (i.e. equation 5) in reality would lead to the high instability of the sampling process. Did you experience any issues of this kind?

2. How did you select the target property values when sampling? Did you randomly sample these values from the distribution taken from the training set?

**Suggestions:**

1. Not sure if background about EDM is necessary in Section 3 – it seems that the work is based on the latent diffusion only.

2. Following question 1 discussed in "Weaknesses", how does the border between chaotic and semantic stage depend on the type of noise schedule? I.e. linear, cosine, polynomial, etc.

---

> ### Author Response · Authors · 2023-11-22
> **Response to Reviewer 7aDk (1/2)**
>
> We appreciate your careful reading and insightful questions. And we seriously address your concerns accordingly; they are extremely helpful to the submission.
>
>
> > Q1: Questions about chaotic and semantic stages, e.g, the border decision under different noise schedules, property checker, and the ablation study without taking into account the chaotic stage.
>
> Thanks for your constructive suggestions! We do believe that the border decision of chaotic and semantic stages is important. To do this, we visualize generated molecules in the range of $t \in [1000,0]$ at 50-step intervals using the pre-trained diffusion model, without defining an objective. We empirically found most generated molecules in $t=400$ are already widely distributed and have certain shapes. Thus, we established 400 as the boundary. For other noise schedules, we can also first visualize the generated process, and decide an appropriate range.
>
>
> The property checker is to evaluate whether the property function of the input (i.e., the molecules generated at $t$) is out of range. If the output value falls outside the property function's range, this value will not be used to guide the direction to prevent instability of the whole process. We provided more details about the property checker on page 5 of the revised manuscript.
>
>
> The main reason that we distinguish chaotic and semantic stages, is that the judgment of the property checker always returns 'not guide' during the chaotic stage. To improve efficiency, we explicitly provide guidance only during the semantic stage. An ablation study was conducted in A.4.1  on two conditioned properties HOMO/Gap. The results indicated that there is no significant difference in guidance, whether the chaotic stage is included or not.
>
> | Guide from | MAE 1 | MAE 2 |
> |------------|-------|-------|
> | 400        | 262   | 489   |
> | 1000       | 271   | 465   |
>
>
>
> > Q2: While one of the main contributions of this work is multi-condition guidance mechanism, in my opinion a more solid experimental validation of the proposed method is required.
>
>
> Thanks for your valuable advice. We truely believe this suggestion is important and we added a serious of experiments.
>
> 1. Combinations of properties considered in Table 2 look a bit strange. Why is dipole moment used in 4 out of 5 considered combinations as Property 2?
>
> This is because we want to fairly compare our model with EEGSDE. They selected three combinations which all contain dipole. So we added a new combination HUMO and LUMO. And following your suggestion, we added two more combinations: LUMO and gap, HOMO and gap. Please see Table 2 in the manuscript.
>
> 2. Besides, correlation 0.22 doesn't seem too high – aren't there better combinations?
>
> Yes. We found LUMO/gap combination has a higher correlation 0.89. We conducted the experiment on LUMO/gap and obtained a superior performance. Please see Table 2 in the manuscript.
>
>
> 3. Combinations gap/LUMO and gap/HOMO seem to be good candidates to illustrate the dependency from Figure 3(a)
>
> Thanks for your suggestion. We totally agree, and we have added these two combinations to Table 2.
>
> 4. It would be interesting to see some experiments with combination from Figure 3(c). Again, gap/LUMO/HOMO seems to be a good candidate for it.
>
> Thanks for your comments! We introduced an additional experiment for y1 = HOMO, y2=LUMO, and y3=Gap, where y1, y2, y3 satisfy the relation in Figure 3(c). We designed two scenarios, one is that y3 is conflicted with y1 and y2, and the other is y3 is consistent with y1 and y2. In the first scenario, the properties HOMO, LUMO and Gap were independently sampled, introducing the possibility of conflicting requirements. Consequently, no molecule could perfectly satisfy the targeted properties, yet MUDM still demonstrated commendable performance. In contrast, the second scenario did not present any inherent conflict among the three properties since Gap is computed from the other two properties. As expected, the performance in this conflict-free setting was superior, as indicated by the lower mean absolute errors (MAEs). These results not only underscore the robustness of MUDM in handling multi-objective tasks with complex inter-property relationships but also highlight its capability to generate feasible molecular structures even in the presence of potential property conflicts.
>
> | Conflict | MAE 1 | MAE 2 | MAE 3 |
> |----------|-------|-------|-------|
> | Yes      | 479   | 545   | 678   |
> | No       | 283   | 392   | 255   |

---

> ### Author Response · Authors · 2023-11-22
> **Response to Reviewer 7aDk (2/2)**
>
> > Q3: "We found it is important to set reasonable guidance weights for multi-condition guidance" – how did authors find these weights and how much do the final results depend on these weights?
>
> Thanks for your questions! To solve your concern we did a new ablation study on the properties LOMO/gap in Appendix A.4.3. It shows how different weights influence the performance. We decided the hyper-parameters by grid search, i.e., search from (1,1), (1,2), (2,1), (3,1), (1,3) on a small batch of generated molecules. Different weights have impacts on the final results. But in most cases, we can simply set all weights as 1.
>
> | Weight 1 | Weight 2 | MAE 1 | MAE 2 |
> |----------|----------|-------|-------|
> | 1        | 1        | 361   | 228   |
> | 1        | 2        | 540   | 456   |
> | 2        | 1        | 393   | 342   |
> | 1        | 3        | 687   | 545   |
> | 3        | 1        | 613   | 526   |
>
>
>
>
> > Q4: As authors discuss in Section 4.1, point estimation used in DPS is very inaccurate, especially in case of 3D molecule generation. It would be interesting to see some experiments with DPS (e.g. for single property conditioning) and comparison with the proposed MC-based method.
>
> Thanks for your suggestions. We did this ablation study on the properties LUMO/gap in Appendix A.4.2 and found the performance of MC sampling is better than DPS.
>
> | Sampling method | MAE 1 | MAE 2 |
> |-----------------|-------|-------|
> | DPS             | 472   | 396   |
> | MC sampling     | 361   | 228   |
>
>
> > Q5: How does the sampling time depend on the number of MC samples? I would be also interested to see the comparison of the sampling time with other methods.
>
> Thanks for your question. The sampling time for a batch of 20 molecules is 77.4 seconds when the number of MC samples is 1, And it is 288.9 seconds as the number of MC samples is equal to 10. The experiments are taken on a single GPU (A100). EEGSDE has roughly the same sampling time. However, it should be noted that EEGSDE needs ~3 days to train a new diffusion model before sampling.
>
>
>
> > Q6: I had an impression that adding another gradient to the denoising process (i.e. equation 5) in reality would lead to the high instability of the sampling process. Did you experience any issues of this kind?
>
> Thanks for your concerns. We found the guidance can be unstable in the early stage during the denoising process. However, we only apply guidance after 400 steps. In this stage, the estimated molecules are more valid and the gradient is relatively stable. In addition, MC sampling method, property checker, as well as time-travel technique are helpful to mitigate this issue.
>
>
> > Q7: How did you select the target property values when sampling? Did you randomly sample these values from the distribution taken from the training set?
>
> Yes, you are absolutely right. It is also the experimental setting used in the baseline papers.
>
>
>
> > Q8: Not sure if background about EDM is necessary in Section 3 – it seems that the work is based on the latent diffusion only.
>
> Thank you for pointing out the relevance of the EDM background in our manuscript. The inclusion of EDM background serves to contextualize the development of diffusion models in the field of 3D molecular generation. By outlining the EDM approach, we aim to make readers easier to understand GEOLDM. Following your suggestion, we compressed this section in the new submission.
>
> We appreciate your careful reading and suggestions for this paper. We believe these revisions and new supplementary experiments help improve the manuscript a lot.

---

### Official Review · Reviewer_LmRX · 2023-10-31

**Soundness:** 3 good
**Presentation:** 3 good
**Contribution:** 3 good
**Rating:** 6
**Confidence:** 4

**Summary:**

In this paper, the authors proposed a new method for 3D molecule generation. Based on the experimental results, their method showed superiority, compared with other existing methods on several real data sets.

**Strengths:**

The author describes the fundamental algorithm well; and they seem to give all relevant information to understand and reproduce their algorithm.

The overall writing is satisfactory. The writing is fluent and clear and the ideas are easy to follow.

The proposed method is relative better than previous methods, which is not lack of significance.

**Weaknesses:**

(1) To make their results more convincing, they should compare their method with more latest state-of-the-art methods.
(2) It remains unclear whether the proposed method is sensitive to the hyper-parameters and how to setup the values in general cases.
(3) Lack of description of the details of the datasets.

**Questions:**

(1) To make their results more convincing, they should compare their method with more latest state-of-the-art methods.
(2) It remains unclear whether the proposed method is sensitive to the hyper-parameters and how to setup the values in general cases.
(3) Lack of description of the details of the datasets.

---

> ### Author Response · Authors · 2023-11-22
> **Response to Reviewer LmRX**
>
> The reviewer is positive about the design of the proposed method, and the empirical results. This is an excellent support to our work. And reviewer encourages us to take more ablation studies to investigate the sensitivity of the hyper-parameters and add more details of the dataset. Here are our responses to your questions.
>
>
> > Q1: To make their results more convincing, they should compare their method with more latest state-of-the-art methods.
>
> A1: Thanks for your suggestions! Our proposed MUDM is designed for training-free multi-objective 3D molecule generation based on off-the-shelf pre-trained diffusion model and time-independent property prediction functions. So MUDM relies on minimal prerequisites. However, our baselines are very strong, where they can conduct additional trainings to enhance the alignment of generated molecules with objectives. One retrained the diffusion model for multiple objectives, and the other trained time-dependent property prediction functions. Despite these alignments, MUDM can still achieve superior performance, which is sufficient to demonstrate its effectiveness.
>
>
> > Q2: It remains unclear whether the proposed method is sensitive to the hyper-parameters and how to setup the values in general cases.
>
> A2:
> Thanks for your comments. The sensitivity of MUDM to hyper-parameters is an essential aspect of our study. To address this, we have conducted comprehensive ablation studies, with detailed results included in the appendix of our revised submission. These studies investigate the influence of different hyper-parameters on the model's performance.
>
> **Ablation Study**:
>
> - **Influence of Guidance Step (Appendix A.4.1)**: We explored how different guidance steps during the diffusion process affect the model’s accuracy. We found that setting guidance step from 400 can achive a balance between the computation and performance.
>
>  - **Influence of Condition Weights (Appendix A.4.3)**: Another critical aspect we examined is the influence of condition weights on multi-objective molecule generation. Our findings indicate that varying these weights allows for nuanced control over the trade-offs between different properties. This is particularly crucial when dealing with conflicting objectives, as it enables the model to prioritize certain properties over others based on the specific requirements of the generation task. In practice, we can simply set all weights as 1.
>
>
>
> > Q3: Lack of description of the details of the datasets
>
> A3: Thanks for this concern! We have added more details of the datasets in Appendix A.6.
>
>
> Following the reviewer's comment improves the quality of the submission significantly. We genuinely appreciate the reviewer's effort and help.

---

### Official Review · Reviewer_yusH · 2023-11-01

**Soundness:** 2 fair
**Presentation:** 3 good
**Contribution:** 3 good
**Rating:** 6
**Confidence:** 2

**Summary:**

This paper presents a training-free conditional 3D molecular generation algorithm based on off-the-shelf unconditional diffusion models to address the issues of efficient, flexible, and potential properties conflict.

**Strengths:**

This paper is well-organized and written. The issue of potential properties conflict and the proposed training-free conditional generation method are both interesting.

**Weaknesses:**

Refer to the question part.

**Questions:**

Could the authors give some insights into why the proposed method can handle the potential properties conflict?

---

> ### Author Response · Authors · 2023-11-22
> **Response to Reviewer yusH**
>
> Thanks for the reviewer's supportive evaluation of our work. We are trying our best to address your concerns with the following answers.
>
> > Q: Could the authors give some insights into why the proposed method can handle the potential properties conflict?
>
> 1. **Modeling Property Relationships as a Probabilistic Graph**: MUDM models the relationships between multiple properties as a probabilistic graph. This allows the method to understand and incorporate the dependencies and potential conflicts between different properties. By doing so, MUDM can adjust the generation process to account for these relationships, thereby effectively managing conflicts.
>
> 2. **Weighted Sum of Gradients for Multiple-Objective Guidance**: In cases of multi-objective tasks, MUDM computes the guidance as a weighted sum of the gradients of each property with respect to the latent variable. This approach takes into account the interdependencies between properties. By considering these dependencies in the calculation, MUDM can navigate through potential conflicts by prioritizing certain properties over others based on their relative importance or the specific requirements of the generation task.
>
> 3. **Flexibility in Handling Conflicting Objectives**: The design of MUDM inherently allows for a degree of flexibility in handling conflicting objectives. This flexibility is evident in how MUDM approaches the generation process, adapting to different scenarios where properties may have varying degrees of conflict or compatibility.
>
> 4. **Experimental Evidence**: The effectiveness of MUDM in handling property conflicts is supported by experimental results. For instance, when examining combinations of quantum properties, MUDM demonstrated superior performance compared to methods like conditional EDM and EEGSDE, even in scenarios with correlated property combinations that could potentially introduce conflicts. We added a new experiment in the revised manuscript A.5. In this experiment, when there are conflicts for the three properties, the proposed framework can still produce molecules with closed targeted properties.
>
> 5. **Advanced Sampling Techniques**: The utilization of advanced techniques like Monte Carlo sampling and the 'time-travel' mechanism during the inference stage contributes to the robustness of MUDM. These techniques ensure more accurate property estimations and help in navigating through the complexities introduced by conflicting properties.
>
> We've made updates based on the feedback provided, and we believe that these changes substantially improve the manuscript. We're grateful for the valuable comments and thank you for your time and expertise.

---

### Author Response · Authors · 2023-11-22
**Revised manuscript uploaded and response to reviewers' comments posted**

Dear Reviewers,

Thanks for your constructive comments and suggestions, which are helpful for improving our manuscript. We have posted the response and revised version of our paper. There are many extra new experiment results added to the paper. We also revised some clarifications and added more experiment details accordingly. We are looking forward to your further feedback. Here we want to highlight several contributions mentioned in the reviews:

- The issue of potential properties conflict and the proposed training-free conditional generation method are both interesting.
- The author describes the fundamental algorithm well; and they give all relevant information to understand and reproduce their algorithm.
- Authors provide a solid mathematical framework for the proposed guidance mechanism.
- Their main contribution is extending the guidance on single molecular property to multiple properties simultaneously by building a probabilistic graph to model the relationships between various properties. The proposed method is novel and interesting.

In conclusion, we have incorporated the insightful feedback provided by the reviewers, enhancing the overall quality of our paper. We sincerely value the time and effort invested in reviewing our work and believe that the revised manuscript and supplementary experiments address the concerns raised.

Best Regards,
Authors

---

### Meta-Review · Area_Chair_gP5r · 2023-12-06

**Metareview:**

This paper introduces a novel algorithm, MUDM, which concerns generating 3D molecular structures with desired properties. The key feature of MUDM is its ability to guide off-the-shelf unconditional diffusion models in a training-free fashion for conditional generation tasks, which demonstrates clear advantages in terms of efficiency, flexibility, and robust handling of potential property conflicts.

Strengths:
- The presentation of the paper is effective, which is clear writing, and  has a well-organized structure.
- The soundness of the proposed method and the contribution to the research field were recognized positively.
- Evidence of MUDM outperforming other state-of-the-art approaches, as shown in the experiments, was considered convincing.

Weaknesses:
- Reviewers found some areas for improvement, particularly in the experimental setup, such as extending comparisons with more up-to-date methods, detailing dataset descriptors, and explicitly addressing the sensitivity to hyperparameters.
- Addig a more detailed exploration of certain technical aspects is helpful, such as the collocation of chaotic and semantic stages and the justification of the property checker utility.

**Justification For Why Not Higher Score:**

N/A

**Justification For Why Not Lower Score:**

In light of the reviewers' assessments and the thorough rebuttal, which led to a revision that considerably improved the submission, the paper  incorporates strong contributions to the field of 3D molecule generation and succeeds in extending beyond existing works.

---

### Decision · Program_Chairs · 2024-01-16

Accept (poster)